# Simulation of Daily Iron Intake by Actual Diet Considering Future Trends in Wheat and Rice Biofortification, Environmental, and Dietary Factors: An Italian Case Study

**DOI:** 10.3390/nu16234097

**Published:** 2024-11-28

**Authors:** Luca Benvenuti, Stefania Sette, Alberto De Santis, Patrizia Riso, Katia Petroni, Cristina Crosatti, Alessia Losa, Deborah Martone, Daniela Martini, Luigi Cattivelli, Marika Ferrari

**Affiliations:** 1Department of Computer, Control and Management Engineering, Sapienza University of Rome, Via Ariosto 25, 00185 Rome, Italy; luca.benvenuti@diag.uniroma1.it (L.B.); desantis@diag.uniroma1.it (A.D.S.); 2Research Centre for Food and Nutrition, Council for Agricultural and Economics Research, Via Ardeatina 546, 00178 Rome, Italy; stefania.sette@crea.gov.it (S.S.); deborah.martone@crea.gov.it (D.M.); 3Department of Food, Environmental and Nutritional Sciences (DeFENS), Università degli Studi di Milano, Via Celoria 2, 20133 Milan, Italy; patrizia.riso@unimi.it (P.R.); daniela.martini@unimi.it (D.M.); 4Department of Biosciences, Università degli Studi di Milano, Via Celoria 26, 20133 Milan, Italy; katia.petroni@unimi.it; 5Research Centre for Genomics and Bioinformatics, Council for Agricultural and Economics Research, Via San Protaso 302, 29017 Fiorenzuola d’Arda, Italy; cristina.crosatti@crea.gov.it (C.C.); luigi.cattivelli@crea.gov.it (L.C.); 6Research Centre for Genomics and Bioinformatics, Council for Agricultural and Economics Research, Via Paullese 28, 26836 Montanaso Lombardo, Italy; alessia.losa@crea.gov.it

**Keywords:** iron adequacy, food consumption, wheat, rice, iron content, carbon dioxide emission

## Abstract

Background and aim: Cereals’ iron content is a major contributor to dietary iron intake in Europe and a potential for biofortification. A simulation of daily iron intake from wheat and rice over the next 20 years will be quantified. Methods: Food items, and energy and iron intake by age classes are estimated using the Italian dietary survey (IV SCAI). Iron intake and adequacy estimation trends were categorized in four scenarios compared to a baseline (basic scenario; only climate change effects): over wheat and rice biofortification effects (scenario 1); over the shift in whole wheat consumption of up to 50% of the total amount of wheat-based foods (scenario 2); over the shift in brown rice consumption up to 100% of the total amount of rice (scenario 3); over the cumulative effects of biofortifications and whole wheat and brown rice consumption (scenario 4). Results: Increasing the iron intake from wheat and rice biofortification and the shift in whole wheat consumption is similar and sufficient to recover the baseline iron depletion effect due to climate change. The shift in brown rice consumption produces a negligible increment in iron intake. The cumulative effects of the corrective actions considered in the scenarios can significantly reduce the iron intake inadequacy, despite not reaching the recommended levels. Conclusions: Corrective actions including biofortification and whole grain consumption are still far from ensuring the full recovery in children and females of fertile age as at-risk groups of iron deficiency. Further actions are needed considering other biofortified food sources, fortified foods, and/or dietary food diversification.

## 1. Introduction

Food is an integral part of human existence, since it contains nutrients and most importantly, micronutrients, namely vitamins and mineral substances essential for health, development, and physical condition [1]. Food consumption represents the main route of intake of all the essential elements for the general population, although the absorption and utilization of essential elements depend on the diet composition and on the individual nutritional status. Iron is an essential mineral with a significant impact on public health as it participates in a wide variety of metabolic processes in the human body, including oxygen transport, the replication of deoxyribonucleic acid (DNA), and neurotransmitter synthesis and function [2]. To meet human iron needs, significant amounts of minerals must be provided by the daily diet to replace the iron that is lost from the body and to meet the growth requirements [3]. The composition of a diet is accepted as influencing the iron bioavailability, as illustrated by the global recommendations made by the WHO/FAO [4] which cover diets with different bioavailability (15%, 12%, 10%, and 5%). However, no clear justification is provided for the selection of these values and, additionally, the effects of dietary enhancers and inhibitors on iron absorption may only be detected in individuals with a higher absorptive capacity as a result of increased iron requirements [5]. It is supposed that the iron status has an even greater effect on iron absorption than diet composition itself, since individuals with a low iron store content tend to absorb more iron from food [6]. A systematic review found large variations in the mean non heme absorption (0.7–22.9%) between studies, which depended on the human iron status as diet was shown to have a greater effect at low serum and plasma ferritin concentrations [6]. Moreover, it should be taken into consideration that other factors may influence the overall iron absorption, including the co-presence of vitamin C which may facilitate absorption by creating a more acidic environment in the stomach. In Europe, the main contributors of dietary iron are cereals and meat [7]. In Italy, about 70% of the iron intake derived from plant-based products, i.e., under the form of non-heme iron in which the principal products are represented by “cereals, cereals products and substitutes” contributed to 31.3% of the total intake (one-third came from the food subgroups of “Bread”), while meat products contributed to 16.9% under the form of heme iron [8]. Considering the transition to sustainable food consumption that includes a reduction in meat portions and an increase in plant-based food sources, an optimized sustainable dietary pattern that has developed in Italy [9] shows an increase in the cereal consumption in both males and females as well as an indirect increase in the iron contribution to the daily diet. On the other hand, higher phytate intake from eating more plant-based products results in low bioavailable iron in modelled diets that tend to limit the overall health benefits of the healthiest dietary patterns [10]. This is particularly important in whole grain foods since the bran in the grain kernel is rich in phytate which may inhibit zinc absorption, despite the consumption of wholegrains being overall undoubtedly associated with improved health [11]. Also, for this reason, the iron intake contribution of cereal products in the last decade has increased due to the iron enrichment of white flour and rice, as well as the research progress in modulating the iron concentration in crops although there have been studies on genetic diversity to breed biofortified wheat and rice and bioavailability [12].

### 1.1. The Iron Grain Concentration in Wheat Genetic Diversity and Future Trends

The concentration of iron in wheat grains depends on genetics determinants, environmental factors (e.g., mineral concentration and bioavailability in the soil, climate conditions, crop management practices), and their interactions. Several studies have suggested wild and primitive accessions of wheat as potential iron-enriched sources, and a general trend towards a reduction in the average iron grain concentration was detected moving from wild to domesticated wheat and from diploid (einkorn, Ae. tauschii) to polyploid (durum and bread wheat) species [13,14,15]. Significant diversity also exists within cultivars, since the literature data for durum and bread wheat report grain iron concentrations in a range from 24 to 65 mg/kg (more frequently between 40 and 50) depending on the genotypes and environmental conditions [16,17,18,19,20]. Furthermore, several studies have suggested that iron grain content is up to 20% lower in modern semidwarf cultivars compared to pre-green revolution genotypes [21,22,23], a finding associated with a dilution effect due to the increased yield of modern wheat varieties [24].

The atmospheric CO_2_ concentration [CO_2_] has progressively increased since the beginning of the industrial revolution, and the CO_2_ concentration is predicted to be between 730 and 1020 ppm by the end of the current century [25]. Since the photosynthetic activity in C3 plants is CO_2_-limited, the most obvious impact of an increased atmospheric CO_2_ concentration is represented by a stimulation of photosynthetic activity (carbon fertilization) that, in turn, results in higher plant biomass. Although the CO_2_ response can be modified by additional stress factors (e.g., drought, nitrogen limitation), most studies carried out in Free Air CO_2_ Enrichment (FACE) facilities indicate in wheat, an increased yield of approximately 10% when the atmospheric [CO_2_] reaches 550–600 ppm [26,27]. Besides this general effect on the plant biomass and yield, some variations in the composition of wheat grain have also been reported and a body of evidence suggests that increased atmospheric [CO_2_] led to a general decrease in the protein and iron concentration in wheat grains.

A FACE experiment carried out in Germany testing eight varieties of bread wheat grown over 3 years (2004–2007) at 550 ppm of CO_2_ reported a decrease in the iron concentration of about 10% [28], and a similar result was reported by a FACE experiment run in Australia [29]. A 25% reduction in the iron grain concentration was instead reported by Beleggia et al. [30] during a FACE experiment run in Italy in the period 2011–2012 with 12 durum wheat genotypes grown at 570 ppm of CO_2_. Besides wheat, similar results have been obtained for other crops, and several meta-analyses of the data for C3 cereals and legumes have called for an early warning concluding that the increasing CO_2_ threatens human nutrition [31,32,33].

The results obtained with plants grown at elevated [CO_2_] are, to some extent, similar to those obtained by comparing varieties selected before and after the introduction of semi-dwarf genotypes. As the yield increases, a general trend towards the reduction in the iron grain concentration can be detected, a trend supporting a dilution effect between starch and minerals probably also explained by the specific sub-localization of Fe in the grains with a maximum concentration in the aleurone layer and embryo of the mature grain [34]. The extent of yields increasing due to an increased grain size is obtained at the expense of iron grain concentration since larger seeds have proportionally less aleurone/embryo tissues.

Overall, different factors might affect the iron concentration in wheat and wheat-based food products. First, there is a general trend towards the reduction in iron in the grains sustained by breeding for increased yields during the last century, often carried out without due attention to the negative correlation between the yield potential and iron grain concentration. Then, this negative trend can strengthen due to the impact of elevated [CO_2_] on crop yields in regions not exposed to drought stress. Overall, if no action is taken, projecting the current trends in the future, the concentration of iron in wheat could decline by 20% by the middle of the century, particularly for wheat grown in temperate regions. Nevertheless, several research actions have clearly demonstrated that it is possible to counteract these negative trends by selecting genetic factors that promote the accumulation of iron in wheat kernels. Specific loci from landraces or wild materials are an attractive option. For instance, the introgression of the Grain Protein Content B1 (GPC-B1) locus from wild emmer into modern wheat accelerates the senescence and leads to up to 18% more iron in the seeds [35]. Alternatively, the selection for yield coupled with the selection for iron seed content is also effective, and [36] have reported a 0.3 mg kg−1 year-1 of iron gain over 11 years of breeding improvement. Plant biotechnology can also act as a powerful tool for increasing the iron content and its bioavailability, with an increased iron seed content of up to 40% [37,38], but regulation in many countries limits the diffusion of genetically modified organisms.

### 1.2. Effect of Elevated CO_2_ on the Iron Content and Rice Biofortification

Elevated CO_2_ combined with heat stress (3–5 °C) significantly decreases the productivity of rice crops by 20–30% in tropical and subtropical regions because of the delayed flowering and seed set and evapotranspiration [39,40,41]. FACE or Open Top Chamber (OTC) studies, comparing ambient CO_2_ (374–425 ppm) versus elevated CO_2_ levels (550–625 ppm), demonstrated that iron is consistently reduced by 8 to 20% in rice cultivars grown in different countries [42,43]. This is expected to especially impact populations, such as those in Asia, mostly relying on rice as their food source [43].

Rice provides insufficient iron to meet the daily requirement, since iron is primarily located in the aleurone and embryo, which are removed in the starch-rich polished grains (endosperm). Furthermore, the iron bioavailability is limited by dietary antinutrients, such as condensed tannins and inositol phosphate, which form very stable complexes with iron and other minerals (i.e., phytates) [44,45].

Considering the daily consumption, food processing, and estimated iron bioavailability in rice, the HarvestPlus consortium has set 15 μg/g dry weight (DW) as the iron content that should be present in polished rice grains, to provide 30% of the dietary estimated average requirement (EAR) of iron [46].

In the last few decades, several techniques have been explored for their potential role in increasing the iron content in food and in turn iron intake and iron status. Among them, a simple strategy can be represented by the fortification by adding iron directly to the food during processing. Besides fortification, biofortification strategies based on both conventional breeding and metabolic engineering techniques have been widely adopted [47,48,49], and their use has been explored for their cost-effectiveness, since it is a one-time investment to develop bio-fortified crops, and the recurrent costs are low, different from fortification techniques [50]. The International Rice Research Institute (IRRI) has developed, through conventional breeding, a biofortified variety of rice with 5-fold more iron (3.21 µg/g DW), after processing and cooking, than local varieties commercially available in the Philippines (0.57 µg/g DW). The daily consumption of this biofortified variety by 192 nonanemic Filipino women for 9 months resulted in a 20% increase in their iron status (i.e., ferritin and body iron) and was, however, estimated to increase the iron intake only from 46% to 56% of the Recommended Daily Allowance (RDA) [51,52].

GWAS and QTL mapping are currently underway, to identify SNPs and the chromosomal regions associated with an increased iron concentration in rice grains. High iron elite lines with Fe- and Zn-enhancing QTLs and twenty elite lines with iron-enhancing QTLs leading to a 10–14 µg/g iron concentration in polished rice grains have been identified [53,54].

The most significant attempt of metabolic engineering obtained a 1.6-fold increase in the iron content in unpolished rice grains (20.53 µg/g DW) and a 9.6-fold increase in polished ones (i.e., 9.6 µg/g DW), which is closer to the 15 µg/g DW target set to provide 30% of EAR [55,56]. However, the consumption of genetically modified biofortified rice varieties is limited by national regulations.

This study aims to quantify in the actual Italian diet and in different age groups the contribution of the cereal food group, in terms of wheat and rice, to the daily iron intake and to compare the total intake to the recommended level by simulating four trend scenarios of iron content in wheat and rice over the next 20 years considering biofortification, environmental, and dietary factors.

## 2. Materials and Methods

### 2.1. Dietary Data

The estimation of the iron intake was calculated by using the data not published of Italian food consumption Survey (IV SCAI 2017–2020) [57,58]. The survey was planned and performed in two separate and concurrent phases according to the population groups: children, from 3 months to 9 years; adults, from 10 to 74 years. Overall, the data were collected for over 800 children aged between 3 months and 9 years, and approximately 1200 adolescents, adults, and older adults aged 10–74 years. The population studied was living in Italy; the sampling unit was the individual and included stratification by sex, age, and the main Italian geographical areas (Northwest, Northeast, Centre, and South and Islands).

All foods, beverages, food supplements, and medicines containing nutrients consumed were assessed on two independent days which were not consecutive (at least 15 days apart) through estimated individual dietary records (from 3 months to 9 years) and a 24 h recall (from 10 to 74 years). Survey days were proportionally distributed for the 4 seasons, 29% on holidays and 71% on weekdays.

The amount of food consumed was estimated using pictures of food portions and household measurements (e.g., glasses, cups, etc.) or, especially for quantifying commercial food portions, the amount was estimated in weight/volume or standard units.

The survey methodology was described in detail by [57,58]. The food items were coded according to the FoodEx2 classification system to provide a standardized database at the European level. Each item was also classified into 15 food categories and 91 subcategories, the latter of which increased from the previous Italian national survey; this was due to considering both baby food and to better quantify the consumption of specific substances, i.e., artificial sweeteners or sauce and condiments or flavour. Individual energy and nutrients intakes were calculated with the use of the in-house software “FoodSoft 1.0” developed by the CREA Research Centre for Food and Nutrition. The software uses a database of food nutrient composition that is continuously updated for new food and supplement formulations on the Italian market.

### 2.2. Wheat and Rice Consumption

To evaluate the consumption of wheat and rice and their contribution to the total iron intake, all the foods containing wheat and/or rice were selected. For each industrial product, such as breakfast cereals, biscuits, breads, crackers, cakes, etc., the label was checked to obtain the amount of wheat or rice contained in the recipe. On this basis, in addition to the quantity, the energy and iron fraction from wheat or rice in each product was calculated. The same process was applied to products such as bread, pasta, and rice prepared at home.

For each individual, the average amount of wheat and rice ingested (average of the two survey days) and their contribution to the average daily energy (kcal/day) and average daily iron intake (mg/day) were estimated. Average total intake, energy and iron intakes, and the contribution of wheat and rice to energy and iron intakesare shown as mean, median, and percentiles of the distribution of intakes by sex and age. The age categories were infants (<1 years), toddlers (1–2 years), children (3–9 years), adolescents (10–17 years), adults (18–64 years), and the elderly (65–74 years). The data of iron intake are also expressed as nutritional density (mg/1000 kcal).

### 2.3. Daily Iron Intake and Adequacy Scenarios

The trend data for iron intake from common wheat and rice over the next 20 years are estimated based on the current dietary pattern for all population age groups, except the youngest (<1 year), and categorized into 5 scenarios. The consumption of wheat-based foods and rice is assumed to remain at the current levels, for both males and females.

Basic scenario: the effect of climate change on iron intake from the actual diet. In the first trend data model, an estimation of the iron intake and adequacy in the middle of the century, resulting in a climate change effect if no corrective action is taken was generated. An iron depletion of both wheat and rice by 20% was considered.

Scenario 1: the effect of biofortification on iron intake from the basic scenario. In the first corrective scenario, the improvement of iron intake formulated from the basic scenario resulting in wheat and grain fortification was calculated. An increase in iron intake of 30% has been estimated for both wheat and rice fortification.

Scenario 2: the effect of the shift in whole wheat consumption to 50% on the iron intake from the basic model. In the second corrective model, the iron intake and adequacy formulated from the basic method resulting in the shift in whole wheat consumption to 50% was calculated.

Scenario 3: the effect of the shift in brown rice consumption to 100% on the iron intake from the basic model. In the third corrective scenario, the iron intake and adequacy formulated from the basic methods, resulting in the shift in whole rice consumption to 100%, was estimated.

Scenario 4: The effect of fortification and shifting whole wheat consumption to 50% and brown rice consumption to 100% on iron intake from the baseline model. In the fourth correction scenario, both the fortification and the shift in whole grain consumption, 50% for wheat and 100% for rice, were considered to estimate the improvement in iron intake and adequacy compared to the baseline scenario

### 2.4. Statistical Analyses

Statistical analysis was performed using the SAS software, version 9.4 (SAS Institute, Inc.; Cary, NC, USA), and Microsoft Excel results are expressed as the median with ranges at the 5th and 95th percentiles range, as well as mean (SD). The Italian Reference Values for iron in the diet considered for this study [59] aim to maintain iron stores by considering the different components that contribute to determining the metabolic iron requirement and taking into account the bioavailability of iron. The reference levels are expressed as the average dietary requirement (AR) and population reference intake (Population Reference Intake, PRI). Then, iron intake analyses were determined using the following PRI cut-off values: 11 mg/day for infants < 1 year; 8 mg/day for children 1–3 years; 11 mg/day for children 4–6 years; 13 mg/day for children 7–10 years; 10 mg/day for adolescent males 11–14 years; 13 mg/day for adolescent males 15–17 years; 10/18 mg/day for adolescent females 11–14 years; 18 mg/day for adolescent females 15–17 years; 10 mg/day for adults and older males (>18 years); 18 mg/day for adult females 18–29 years; 18/10 mg/day for adult females 30–59 years; 10 mg/day for adults and older (>60 years). The Kruskal–Wallis test was used to compare the mean intake values of the wheat and rice variables.

## 3. Results

### 3.1. Dietary Assessment

The median intake of food and beverages consumed by the population was 2.291 g/day (mean 2.370 g/day) which corresponds to 1.583 kcal/day of energy intake (mean 1.659 kcal/day) (Table 1). The median dietary iron intake in males was 9.6 mg/day and ranged from 5.8 (age 1–2 y) to 12.8 mg/day (age 65–74 y) (with the mean from 6.3 (age < 1 y) to 14.2 mg/day). In females, the median intake was 8.3 mg/day and ranged from 5.3 to 10.6 mg/day (age 65–74 y) (mean from 5.0 to10.3 mg/day) (Table 1) with the iron intake increasing with age in both males and females. The 95th percentile iron intake was 18.7 mg/day and 15.2 mg/day for males and females, respectively, while the 5th percentile was 3.9 mg/day and 3.5 mg/day, respectively. Compared with the reference values, the mean iron intake was lower than the recommendations for all the age groups considered in this study, except for the elderly group (65–74 years old), where the value was higher (11.9 mg/day) and therefore adequate. In the children’s groups (<1 year, 1–2 years, 3–9 years) and adolescents (10–17 years), the average iron intake was lower than the recommendation (PRI) with a gap of −0.2–4.6 mg/day and −1.9–8.0 mg/day for males and females, respectively. In the adult group (18–64 years), the average iron intake below the reference value was observed in females (−6.6 mg/day), while the values for males were adequate according to the recommendations for this age group.

In Table 2a,b, the differences by age group are statistically significant for all the variables (*p* < 0.001) for the whole sample. The differences between males and females are highly significant for the wheat variables (quantities, energy, and iron intake) (*p* < 0.001) (Table 2a) but not for the rice variables (Table 2b).

The mean intake of wheat was the highest in the elderly group (65–74 years) in both males (156.7 g/day) and females (121.1 g/day), while it was the lowest in children (<1 years) in both males (29.3 g/day) and females (29.5 g/day) (Table 2). For rice, the average intake was highest for adult males (18-64 years) (33.3 g/day) and lowest for children (<1 year) and females. For the total sample, the daily dietary energy intake was 352.6 kcal from wheat (Table 2), much higher than that of rice (9 kcal) (Table 3) with a contribution to total energy intake of 21.2% and 0.5% for wheat and rice, respectively. The median energy intake from wheat and rice consumption (Table 2 and Table 3) ranged from 71.3 (age 3–9 y) to 523.8 kcal/day (age 10–17 y) and 0.9 (age < 1 y) to 59.5 kcal/day (age 10–64 y), respectively, for males and from 82 (age < 1 y) to 411.2 kcal (age 10–17 y) and 0 (age < 1 y) to 47.4 kcal (age 3–9 y), for females. Overall, a higher median energy intake for wheat in adolescents and for rice in adults for both males and females (10–17 years) was observed. The median iron intake from wheat consumption ranged from 0.4 (age < 1 y) to 1.8 mg/day (ages 10–17 and 65–74 y) and from 0.3 (age < 1 y) to 1.3 mg/day (ages 10–17 and 65–74 y), for males and females, respectively, with the highest median iron intake in adolescents and adults in both males and females.

The mean total cereal intake in all the samples was 199.6 g/day, and 224 and 176.9 in all males and all females, respectively. The median intake ranged from 46.3 to 302.6 g/day for males and 49.5 to 227.3 g/day for females in the different age groups (Table 3). The percentage of wheat intake from the total cereal consumption was the highest in the elderly group in both males and females (63.1% and 61.4%, respectively), while the percentage of rice from the total cereal consumption was the highest in adults for males (18–64 years) (11.9%) and in children (1–2 years) (14.0%) in females.

### 3.2. Projected Changes in the Iron Intake and Adequacy

The analysis of all the scenarios was based on the hypothesis that the consumption of wheat-based foods and rice will remain at current levels. A basic scenario in estimating the iron intake and adequacy (the difference between the actual and the required iron intake level in the middle of the century from now because of climate change effects) was considered. To this end, a cumulative 20% iron depletion of both wheat and rice was considered. The resulting adequacy of iron intake (for males and females) is reported as the baseline in Figure 1, Figure 2, Figure 3 and Figure 4 to compare the mitigation of the effects of climate change according to four different corrective actions. In scenario 1, the iron intake increase resulting from 30% iron biofortification of wheat and rice, with respect to the baseline, was considered. Figure 1 shows a general improvement in the iron adequacy; in particular, the male age group of 10–17 fully recover the adequacy.

Scenario 2 considers a different corrective action consisting of a shift in whole wheat consumption to 50% of the total wheat-based food amount. Figure 2 shows that the results from this action are comparable to those of scenario 1.

In scenario 3, mitigation is obtained by a shift in the whole rice consumption to 100% of the total rice amount. Figure 3 shows that this action has negligible results, by an order of magnitude lower than the previous two scenarios.

The last scenario evaluates the cumulative effect of all the previous corrective actions considered, that is, the biofortification of wheat and rice and shifts in whole-wheat-based food and rice consumptions. The results are shown in Figure 4.

Notably, the corrective actions reduce the iron intake inadequacy despite the depletion due to climate changes.

To assess the statistical significance of the increase in iron intake the 95% confidence interval of the mean daily iron intake was computed from data of each age group of interest, both for males and females. Any increase that is larger than the interval half width is significant with a p.v. lower than 0.025. Table 4 shows that the cumulative increase in the average daily iron intake due to policies defining scenario 4 is significant in all the considered age groups.

## 4. Discussion

The results of the present study report that the current Italian food consumption is critical to ensure the adequate iron intake in vulnerable population groups. Dietary iron intakes by children and adolescents were low compared to the age- and sex-specific nutrient requirements [59] at a time when the physiological demand to support growth and development are the highest [5]. In females, the average dietary iron intake was also lower than the recommendation in the adult group when it is critical for females of fertile age to be adequately prepared for pregnancy with a sufficient iron intake to build up stores [60]. Adult males are at the limits of iron adequacy contrary to what was reported in a study that found, in Europe, 75–87% of dietary iron intake above 9 mg/day in the same population group [61].

In many age groups, the current mean iron intake is lower than the recommendations; nevertheless, the situation is expected to become worse in the near future considering that in Italy, the most significant contribution of iron to the diet comes from cereals [62] and that the increase in atmospheric [CO_2_] [31,32,33] as well as the general breeding trend [21,22,23,24] contribute to a reduction in the iron content. Meyers et al. (2014) demonstrated significant losses of iron in wheat and rice in crops grown in open fields under elevated atmospheric carbon dioxide conditions. It has been suggested that a large number of females in their childbearing years and children under 5 years of age would be at high risk from the CO_2_-mediated iron loss in food crops [63].

On the other hand, there are increasing movements towards food systems and patterns in favour of promoting health and environmental sustainability that encourage a shift towards plant-based diets for human planetary health benefits through the consumption of different plant foods and minimal amounts of red and processed meat. Plant-based diets, if not appropriately balanced, can result in micronutrient deficiencies, such as iron deficiency [64]. Due to the long-lasting effect of this context on nutrition and health where iron is considered a key critical nutrient for future nutritional security, measures to mitigate this impact are urgently needed.

The scenario analyses in this study show how a shift of up to 50% to whole grain (Scenario 2) would lead to a partial recovery in iron intake for all the population groups considered. The benefit of this shift is reported to be higher in males with a peak of up to 0.8 mg/day for adolescents and a recovery of iron adequacy of up to 0.14 mg/day (Appendix A). Wholegrain foods are being actively promoted as part of a healthy, sustainable diet profile, based on the need for higher intakes of plant-based dietary fiber-containing foods and lower consumption of meat and fattier animal products [65,66]. The evidence from scientific literature supports the benefits of dietary grain intake in the prevention of type 2 diabetes, cardiovascular disease, and colorectal, pancreatic, and gastric cancers and suggests the consumption of two to three servings per day (~45 g) of whole grains to achieve a public health goal [67]. In Italy, a study reveals a very low whole grain intake especially in children and adolescents, and the authors suggest public health strategies to increase the whole grain consumption [68]. This low consumption may be attributed to several barriers, including poor availability, lack of appeal, and the cost of wholegrain foods, as well as the difficulty in identifying these products and limited knowledge of their benefits. These factors may restrict the choice of such foods and, in turn, hinder the achievement of this scenario, especially among younger age groups [69]. Since whole grains have a substantial content of phytate that inhibits iron absorption [70], it is recommended to model the dietary patterns to optimize the iron availability depending on the nutritional components that act as enhancers or inhibitors of iron absorption [71] and incorporating a diverse range of foods containing iron and iron-fortified products within a balanced diet [72]. Fortification is considered as a climate-friendly way of delivering micronutrients, as it requires no new agricultural land or infrastructure for crop production and distribution [73].

Likewise, cereal-based foods are commonly used as vehicles for iron biofortification [74], and research involving the application of grain biofortification has shown some success in alleviating the deficiencies in populations unable to achieve diversified food-consumption patterns [75]. The scenario 1 analyses report the improvement of iron intake resulting from 30% iron biofortification of wheat and rice with respect to the baseline, leading to a further recovery of iron intake for all the population with a peak of up to about 0.7 mg/day (Appendix A). This analysis indicates full recovery of mild inadequacy in male adolescents and a small part of recovery for children, but it is still far from indicating full recovery for adolescents and adult females, a population group at a high risk of iron deficiency, today and in the future. A meta-analysis showed that iron-biofortified crop interventions significantly improved the cognitive performance in the attention and memory domains, compared with conventional crops, but there were no significant effects on the categorical outcomes such as iron deficiency or anaemia [76]. This feeding is also reported by another study as a review [77] with the consensus to carry out other interventions to determine the efficacy of iron-biofortified staple crops in relation to human health, including additional functional outcomes and other high-risk populations. In this study, the evaluation of both scenarios 1 and 2 denotes that iron loss due to an increase in CO_2_ and breeding trends can be completely recovered by biofortification (increase range from 0.3 to 0.7 mg/day) or shift to 50% of whole wheat consumption (increase range from 0.2 to 0.8 mg/day) maintaining constant actual cereal amounts.

The iron intake benefits of switching from a rice-based diet to a brown rice-based diet (scenario 3) is lower than those of wheat. In this case, females appear to benefit more than males in all the age groups with iron inadequacy. Despite a 100% shift away from rice, the increase in iron intake from rice is much less than that from wheat, ranging from a half to a fifth of that from wheat. On the other hand, rice represents a lower source of iron in the national food consumption than wheat; this is expected as rice is not a staple food like wheat is in Italy. Although brown rice contains bioactive compounds and micronutrients, including polyphenols, minerals, and vitamins, which are not present in white rice after polishing [44], white rice is more widely consumed than brown rice. The cumulative effect of all the previous corrective actions considered (scenario 4) are able to completely cover the recovery of iron adequacy in adolescent males of up to 1.85 mg/day but still cannot have a complete effect for children and females. For these population groups, other strategies of intervention should be considered such as nutritional education with food diversification that seems to lead to the achievement of an increased intake of iron-rich foods and in the development of an optimal diet composition for the iron bioavailability [77]. Also considering other biofortified food sources and/or fortified foods consumed in sufficient quantities by these target groups could represent a key strategy.

Overall, in the scenario analyses presented in this study, we considered no change in wheat and rice consumption for the middle of the century. The sensitivity of the calculated outcomes (for example, changing food habits or food cost) is straightforward as it is a simple projection; for example, if wheat consumption falls by 10%, on average, 10% from all the projections made is lost.

Our study has certain strengths and limitations. A major strength is the use of the individual data of the last national food consumption survey (IV SCAI 2017–2020) [57,58] which covers all four main geographical areas and all classes of age. In addition, IV SCAI respects the methodology recommended by the European Food Safety Authority [78]. Furthermore, IV SCAI estimates the intake of grain and rice using the brands and labels of products’ data consumed at the specific time of the dietary survey. However, the IV SCAI study, like all studies assessing dietary intakes, is based primarily on self-reporting by the participants, which makes the reliability of the data partly dependent on the cognitive individual abilities of the participants, and on the possible biases in the reporting. In addition, the effect of over- and under-reporting was not taken into consideration in the present analysis, which may have resulted in an over- or under-estimation of whole grain intakes. Furthermore, the predictions on the impact of climate change and of biofortification on the iron grain content in rice and wheat were estimated based on general genetic trends and on a few varieties tested under elevated [CO_2_], without considering the variability in the soil composition and the impact of possible iron fertilization. Finally, we considered only iron from cereal-based foods as a case study, even though iron intake, and in turn, iron status, may be influenced by many factors (e.g., the co-presence of phytate, co-intake of vitamin C) and that many other foods can be a source of iron, the amount of which can also be affected by climate changes.

## 5. Conclusions

Considering the inadequate levels of iron intake in the current diet from the IV SCAI survey data, children and females of fertile age represent the groups at risk in the Italian population for future iron deficiency considering also a worsening of the iron content in wheat and rice due to climate change. Corrective actions using biofortification and whole grain consumption imaging of future scenarios are still far from ensuring full recovery and denote different effects. The biofortification and the shift of 50% in whole wheat consumption have comparable effects; the combination of the two corrective scenarios has a significant effect, but this is not sufficient for the full iron recovery. On the contrary, the 100% shift in brown rice consumption has had an almost negligible effect, i.e., an order of magnitude smaller compared to whole wheat, due to the limited rice consumption in the population considered.

To counteract the effects of climate change on the cereal iron content, the study findings support further corrective actions to improve the iron adequacy beyond the consumption of biofortified and whole grain wheat as well as considering other biofortified food sources, fortified foods, and/or dietary food diversification.

## Figures and Tables

**Figure 1 nutrients-16-04097-f001:**
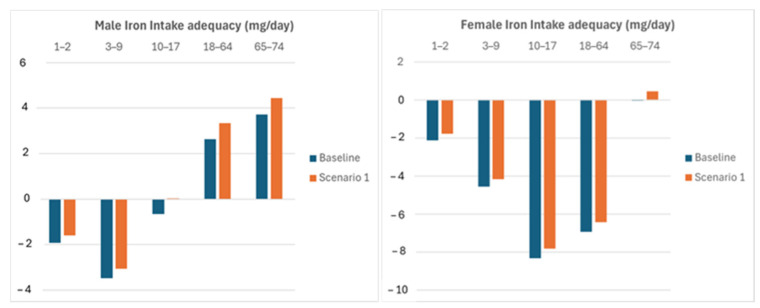
Iron intake adequacy improvement due to iron bio-fortification with respect to the baseline. Zero line represents that iron intake equals the recommended level; negative values denote inadequacy, and positive values denote excess of iron intake.

**Figure 2 nutrients-16-04097-f002:**
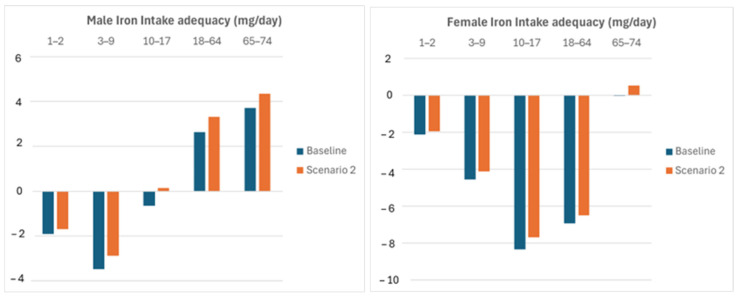
Iron intake adequacy improvement due to a shift in whole-wheat-based food consumption.

**Figure 3 nutrients-16-04097-f003:**
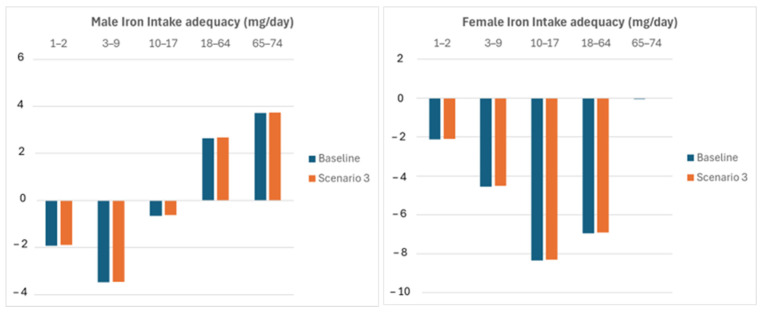
Iron intake adequacy improvement due to a shift in brown rice consumption.

**Figure 4 nutrients-16-04097-f004:**
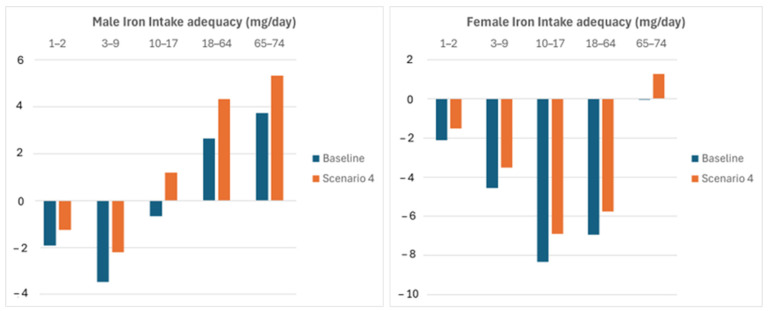
Iron intake adequacy improvement due to bio-fortification of wheat and rice and shifts in whole-wheat-based foods and rice consumptions.

**Table 1 nutrients-16-04097-t001:** Daily food and beverages ingested, daily energy and iron intake, and higher/lower average iron intake than the recommended requirement in the Italian sample of national food consumption data—IV SCAI ^a^.

		Total Amount of Food and Beverages Ingested		Energy Intake		Iron Intake		Average Negative/Positive Iron Intake to Achieve the Recommended Level	
		g/day		kcal/day		mg/day		mg/day	
	N.	Mean (SD)	Median (P5–P95)	Mean (SD)	Median (P5–P95)	Mean (SD)	Median (P5–P95)	Mean Range (SD)	Median Range (P5–P95)
M *	954	2480(1004)	2450 (1031–4212)	1824 (706)	1794 (787–3067)	10.3 (4.7)	9.6 (3.9–18.7)	0.1 (4.7)	−0.5 (−6.5/8.5)
<1	75	1208 (291)	1160 (733–1730)	892 (243)	896 (590–1317)	6.4 (3.3)	6.7 (0.7–12.2)	−4.6 (3.3)	−4.3 (−10.3/1.2)
1–2	162	1500 (443)	1425 (902–2235)	1133 (273)	1099 (703–1623)	6.3 (2.6)	5.8 (2.8–11.0)	−1.7 (2.6)	−2.2 (−5.2/3.0)
3–9	168	2011 (582)	1999 (1049–3042)	1545 (410)	1516 (926–2313)	7.8 (2.7)	7.6 (3.7–14.0)	−3.2 (2.8)	−3.3 (−7.2/2.2)
10–17	138	2971 (780)	2912 (1763–4184)	2218 (573)	2215 (1284–3183)	11.4 (3.7)	11.3 (6.1–17.3)	−0.2 (3.8)	−0.5 (−6.3/6.4)
18–64	346	3150 (820)	3075 (1897–4556)	2270 (563)	2198 (1388–3271)	13.1 (4.1)	12.6 (7.5–20.6)	3.1 (4.1)	2.6 (−2.5/10.6)
65–74	65	3010 (715)	2970 (1838–4478)	2135 (619)	2044 (1403–3337)	14.2 (4.6)	12.8 (8.1–23.4)	4.2 (4.6)	2.8 (−1.9/13.4)
F *	1015	2267 (859)	2190 (983–3814)	1504 (477)	1480 (765–2338)	8.6 (3.7)	8.3 (3.5–15.2)	−5 (4.3)	−5.2 (−11.7/2.6)
<1	75	1131 (341)	1001 (747–1747)	838 (235)	809 (539–1272)	5.0 (3.5)	5.3 (0.2–11.5)	−6 (3.5)	−5.7 (−10.8/0.5)
1–2	160	1509 (396)	1511 (902–2381)	1084 (241)	1094 (733–1503)	6.1 (2.7)	5.6 (2.7–11.9)	−1.9 (2.7)	2.4 (−5.3/3.9)
3–9	171	1828 (488)	1767 (1111–2742)	1411 (327)	1417 (904–1912)	7.1 (2.2)	6.6 (4.2–10.9)	−4.3 (2.5)	−4.7 (−7.7/0.2)
10–17	138	2701 (805)	2574 (1522–3942)	1779(463)	1730 (1132–2538)	9.8 (3.2)	9.7 (5.1–16)	−8 (3.3)	−8.2 (−12.9/−1.8)
18–64	380	2744 (745)	2640 (1679–4153)	1718 (402)	1662 (1160–2412)	10.3 (3.5)	9.8 (5.6–16.7)	−6.6 (4.4)	−7.5 (−12.3/2.3)
65–74	91	2668 (553)	2637 (1769–3557)	1630 (429)	1540 (943–2343)	10.3 (3.1)	10.6 (5.1–15.6)	0.3 (3.1)	0.6 (−4.9/5.6)
All	1969	2370 (938)	2291 (990–4005)	1659 (619)	1583 (769–2751)	9.4 (4.3)	8.9 (3.7–17.1)	−2.5 (5.2)	−2.8 (−10.8/6.4)
<1	150	1170 (318)	1106 (745–1747)	866 (240)	834 (542–1303)	5.7 (3.5)	6.1 (0.3–11.7)	−5.3 (3.5)	−4.9 (−10.7/0.7)
1–2	322	1505 (419)	1457 (902–2245)	1109 (258)	1099 (703–1543)	6.2 (2.6)	5.7 (2.8–11.9)	−1.8 (2.6)	−2.3 (−5.2/3.9)
3–9	339	1918 (544)	1863 (1079–2883)	1477 (376)	1460 (926–2094)	7.4 (2.5)	7 (3.9–11.8)	−3.8 (2.7)	−4.2 (−7.5/1.3)
10–17	276	2836 (803)	2698 (1721–4184)	1999 (565)	1960 (1189–3074)	10.6 (3.6)	9.9 (5.5–16.6)	−4.1 (5.3)	−4.1 (−11.8/5.6)
18–64	726	2935 (807)	2865 (1759–4405)	1979 (558)	1929 (1195–3004)	11.6 (4)	11.1 (6.1–18.8)	−2.0 (6.5)	−1.9 (−11.6/8.5)
65–74	156	2811 (645)	2716 (1835–4048)	1841 (570)	1728 (1011–2777)	11.9 (4.2)	11.5 (5.3–20.4)	1.9 (4.2)	1.5 (−4.7/10.4)

M = male, F = female; SD = standard deviation; P5 = 5° percentile; P95 = 95° percentile. * Years. ^a^ ad hoc elaboration of IV SCAI data not published.

**Table 2 nutrients-16-04097-t002:** a—Daily wheat consumption expressed as quantity, energy, and iron intake in the Italian sample of national food consumption data—IV SCAI ^a^ b—Daily rice consumption expressed as quantity, energy, and iron intake in the Italian sample of national food consumption data—IV SCAI ^a^.

**(a)**							
		**WHEAT INTAKE**					
		**Quantity (g/day)**		**Energy (kcal/day)**		**Iron (mg/day)**	
**Years**	**N**	**Mean (SD)**	**Median (P5–P95)**	**Mean (SD)**	**Median (P5–P95)**	**Mean (SD)**	**Median (P5–P95)**
Males	954	120.2 (78.9)	108.3 (16.8–262.6)	396.2 (246.6)	362 (57.9–861.3)	1.6 (1.3)	1.3 (0.2–4.1)
<1	75	29.3 (28.9)	22.2 (0–82.9)	103 (102.8)	71.3 (0–296.9)	0.8 (1)	0.4 (0–2.9)
1–2	162	66.7 (33)	62.3 (21.6–124.4)	228.6 (111.1)	224.8 (76–439.8)	1 (0.6)	0.9 (0.2–2.1)
3–9	168	102.5 (51.8)	97.9 (27.8–187.5)	354.7 (175.4)	348.5 (93.3–640.1)	1.2 (0.8)	1.1 (0.3–2.8)
10–17	138	163.1 (78.1)	145.8 (57.1–322.7)	545.4 (246.5)	523.8 (194.4–1001.3)	2.1 (1.3)	1.8 (0.7–5)
18–64	346	149.8 (80.9)	140.6 (45.7–316.3)	482.3 (246.6)	453.5 (159–975.1)	2 (1.4)	1.7 (0.5–4.4)
65–74	65	156.7 (76.1)	146.1 (48.4–315.7)	484.8 (225.3)	437.2 (182–948.6)	2.2 (1.5)	1.8 (0.4–5.6)
Females	1015	94.5 (57.9)	86.2 (15.5–196.8)	311.9 (183.3)	290.5 (54.1–636.1)	1.3 (1)	1.1 (0.2–3.5)
<1	75	29.5 (29.5)	22.8 (0–93.8)	101.9 (103)	82 (0–314.1)	0.8 (1.1)	0.3 (0–3.4)
1–2	160	63.8 (29.2)	58.9 (22.1–113.9)	214.8 (96.7)	193 (74.4–384.7)	1 (0.7)	0.8 (0.3–2.5)
3–9	171	86 (41.8)	79.5 (31.4–162.5)	300.2 (146.5)	289.1 (110–580.1)	1.1 (0.7)	0.9 (0.3–2.5)
10–17	138	121.1 (58.3)	116 (39.8–223.8)	407.9 (181.8)	411.2 (136.7–728.6)	1.5 (1)	1.3 (0.4–3.7)
18–64	380	107.8 (60.6)	100.4 (21–204.8)	349.1 (191.4)	331.8 (67.9–670.9)	1.5 (1.1)	1.2 (0.3–3.9)
65–74	91	121.1 (61)	113.7 (33.7–231.7)	371.3 (177.2)	365.2 (107.3–756.1)	1.6 (1.2)	1.3 (0.3–4.3)
All	1969	107 (70)	95 (16.2–238.1)	352.6 (220.3)	320.9 (56.4–763)	1.5 (1.2)	1.2 (0.2–3.8)
<1	150	29.4 (29.1)	22.5 (0–85.6)	102.5 (102.5)	78.8 (0–314.1)	0.8 (1)	0.4 (0–3)
1–2	322	65.3 (31.2)	60.7 (21.7–122.1)	221.7 (104.2)	209.5 (76–417.3)	1 (0.7)	0.8 (0.3–2.2)
3–9	339	94.2 (47.7)	86.5 (30.3–181.1)	327.2 (163.5)	305.3 (104.3–602.2)	1.2 (0.8)	1 (0.3–2.6)
10–17	276	142.1 (72)	134.4 (42.2–263.5)	476.7 (227.1)	446.2 (154.4–903.5)	1.8 (1.2)	1.5 (0.5–4.3)
18–64	726	127.7 (74)	114.6 (28.2–260.6)	412 (229.2)	377 (90.9–827.2)	1.7 (1.3)	1.4 (0.3–4.2)
65–74	156	135.9 (69.6)	121.8 (35–270.5)	418.5 (205.4)	387.5 (116.1–839.8)	1.8 (1.3)	1.5 (0.4–4.4)
**(b)**							
		**RICE INTAKE**					
		**Quantity (g/day)**		**Energy (kcal/day)**		**Iron (mg/day)**	
**Years**	**N**	**Mean (SD)**	**Median (P5/P95)**	**Mean (SD)**	**Median (P5/P95)**	**Mean (SD)**	**Median (P5/P95)**
Males	954	23.5 (38.7)	9.4 (0–86.6)	64.5 (92)	30 (0–230.8)	0.2 (0.3)	0.1 (0–0.8)
<1	75	4.7 (7.2)	0.3 (0–23)	15.8 (24.2)	0.9 (0–81.3)	0.2 (0.4)	0 (0–1.3)
1–2	162	12.9 (25)	7.2 (0–44.7)	33.2 (40.1)	20.1 (0–106.8)	0.1 (0.2)	0.1 (0–0.5)
3–9	168	18.3 (30.8)	9.4 (0–50.9)	53.3 (76.6)	30 (0–171.4)	0.2 (0.3)	0.1 (0–0.6)
10–17	138	30.5 (52.8)	20.3 (0–87.6)	78.3 (86.6)	50.3 (0–246.9)	0.2 (0.3)	0.1 (0–0.8)
18–64	346	33.3 (43.3)	19.9 (0–104)	91.1 (117.4)	59.5 (0–301.7)	0.3 (0.4)	0.2 (0–0.9)
65–74	65	18.2 (25.2)	5 (0–76.2)	57.1 (75.9)	14.9 (0–258.4)	0.2 (0.3)	0 (0–1.2)
Females	1015	20.5 (35.8)	8.1 (0–76.7)	55.2 (75.0)	25.3 (0–214.7)	0.2 (0.3)	0.1 (0–0.7)
<1	75	2.7 (5.2)	0 (0–18)	9.1 (17.1)	0 (0–41.7)	0.1 (0.3)	0 (0–0.6)
1–2	160	15.9 (54)	3.1 (0–40.2)	30 (49.5)	8.9 (0–121.1)	0.1 (0.2)	0 (0–0.5)
3–9	171	22.6 (28.7)	14.7 (0–77.4)	66 (74.4)	47.4 (0–206.3)	0.2 (0.3)	0.1 (0–0.6)
10–17	138	21.3 (26.5)	10.9 (0–73.7)	61.4 (78)	35.5 (0–237.1)	0.2 (0.3)	0.1 (0–0.6)
18–64	380	25.7 (37)	13.6 (0–96.9)	70.7 (87.4)	36.8 (0–243.7)	0.2 (0.4)	0.1 (0–0.9)
65–74	91	15.1 (20.8)	6.8 (0–56.6)	40.3 (49.7)	12.5 (0–132)	0.1 (0.2)	0 (0–0.5)
All	1969	22.0 (37.2)	9 (0–81.9)	59.7 (83.8)	26.8 (0–222.6)	0.2 (0.3)	0.1 (0–0.8)
<1	150	3.7 (6.3)	0 (0–18.7)	12.5 (21.2)	0 (0–67.7)	0.1 (0.4)	0 (0–0.8)
1–2	322	14.4 (41.9)	5.7 (0–40.2)	31.6 (45.0)	15.8 (0–108.8)	0.1 (0.2)	0 (0–0.5)
3–9	339	20.5 (29.8)	11.4 (0–57.9)	59.7 (75.7)	39.5 (0–203.5)	0.2 (0.3)	0.1 (0–0.6)
10–17	276	25.9 (42)	14 (0–83.4)	69.9 (82.7)	44.3 (0–246.9)	0.2 (0.3)	0.1 (0–0.6)
18–64	726	29.3 (40.3)	16.4 (0–100.2)	80.3 (103.2)	47.5 (0–255.7)	0.2 (0.4)	0.1 (0–0.9)
65–74	156	16.4 (22.7)	5.4 (0–56.6)	47.3 (62.3)	14.9 (0–165.6)	0.1 (0.2)	0 (0–0.7)

SD = standard deviation; P5 = 5 percentile; P95 = 95 percentile. ^a^ ad hoc elaboration of IV SCAI data not published.

**Table 3 nutrients-16-04097-t003:** Total cereal intake (g/day) and related contribution (%) of wheat and rice intake in the Italian sample of national food consumption data—IV SCAI ^a^.

		Quantity Total Cereal (g/day)		Quantity Wheat (g/day)	Quantity Rice (g/day)
Years	N.	Mean (SD)	Median (P5/P95)	% of Total Cereal	% of Total Cereal
Males	954	224 (118.8)	212.1 (48.9–432.6)	53.9	10.5
<1	75	54.3 (30.7)	46.3 (9–112)	54.0	8.7
1–2	162	121.5 (51.7)	116.1 (48.2–214.2)	54.9	10.6
3–9	168	194 (76.6)	193.2 (79.9–336.4)	52.8	9.4
10–17	138	304.2 (116.6)	302.6 (135.9–471.4)	53.6	10.0
18–64	346	280.4 (103.8)	264.7 (121.9–476.8)	53.4	11.9
65–74	65	248.4 (93)	226 (124.2–401.3)	63.1	7.3
Females	1015	176.9 (87.4)	169.6 (48–331)	53.4	11.6
<1	75	51.3 (32)	49.5 (6.5–103)	57.5	5.3
1–2	160	113.7 (46.2)	107 (41.4–196)	56.1	14.0
3–9	171	170.4 (63.2)	162.8 (80.6–275.3)	50.5	13.3
10–17	138	231 (83.5)	227.3 (95.2–368.5)	52.4	9.2
18–64	380	202.1 (84.2)	195.4 (78.5–347.3)	53.3	12.7
65–74	91	197.2 (83.2)	198.1 (72.1–353.4)	61.4	7.7
All	1969	199.6 (106.4)	189.1 (48.2–389.3)	53.6	11.0
<1	150	52.8 (31.3)	48.9 (9–106)	55.7	7.0
1–2	322	117.6 (49.1)	112.3 (44.5–205.6)	55.5	12.2
3–9	339	182.1 (71.1)	177.5 (80.6–316.3)	51.7	11.3
10–17	276	267.6 (107.7)	258.4 (111.5–436.9)	53.1	9.7
18–64	726	239.1 (101.9)	225.7 (99.5–437.1)	53.4	12.3
65–74	156	218.5 (90.5)	208 (92.7–387.9)	62.2	7.5

SD = standard deviation; P5 = 5° percentile; P95 = 95° percentile. ^a^ ad hoc elaboration of IV SCAI data not published.

**Table 4 nutrients-16-04097-t004:** Comparison of the cumulative increase in the average daily iron intake obtained with policies of scenario 4, with threshold levels for significance.

Males			Females		
Age Groups (Years)	Least Significant Delta (mg/day)	Delta Scenario 4 (mg/day)	Age Groups (Years)	Least Significant Delta (mg/day)	Delta Scenario 4 (mg/day)
1–2	0.40	0.66	1–2	0.42	0.60
3–9	0.41	1.28	3–9	0.33	1.04
10–17	0.62	1.85	10–17	0.53	1.44
18–64	0.43	1.68	18–64	0.35	1.18
65–74	1.17	1.61	65–74	0.67	1.30

## Data Availability

The original contributions presented in the study are included in the article/Appendix A, further inquiries can be directed to the corresponding author.

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
