# Peer review of "Simulation of Daily Iron Intake by Actual Diet Considering Future Trends in Wheat and Rice Biofortification, Environmental, and Dietary Factors: An Italian Case Study"

_nutrients, 2024, doi:10.3390/nu16234097_

Round 1

Reviewer 1 Report

Comments and Suggestions for Authors

The well-written and easy-to-read manuscript predicts daily iron intake in the Italian diet based on its content of wheat- and rice-based food using data from IV SCAI.   The results in general from the simulations are not in any manner surprising, but the details are interesting. I only have minor comments:

Some words were previously at the end of a line and are not in the present form so that extraneous hyphens are present, e.g. "se-lection".

Some numbers have hyphens rather than periods as decimal points in the text and in Table A1.

Some compounds sentences lack commas, making them run-on sentences.

Please change "/die" to "/day" in the text.  Is "/day" in the tables.

Do not use first person voice, e.g. "we", except in introduction and conclusion.

The text in the figures is difficult to read and needs to be in a darker font.

Author Response

Comment 1: The well-written and easy-to-read manuscript predicts daily iron intake in the Italian diet based on its content of wheat- and rice-based food using data from IV SCAI.   The results in general from the simulations are not in any manner surprising, but the details are interesting. I only have minor comments:

Response 1: We thank the reviewer for the positive comments about our work. We have replied point-by-point to the comments arisen, as described below.

Comment 2: Some words were previously at the end of a line and are not in the present form so that extraneous hyphens are present, e.g. "se-lection".

Response 2: Thank you for noting them, we have properly revised.

Comment 3: Some numbers have hyphens rather than periods as decimal points in the text and in Table A1.

Response 3: according to the reviewer’s comment, we have revised.

Comment 4: Some compounds sentences lack commas, making them run-on sentences.

Response 4: we have rephrased commas in several sentences to shorten them and increase the ease of reading.

Comment 5: Please change "/die" to "/day" in the text.  Is "/day" in the tables.

Response 5: Done as suggested

Comment 6: Do not use first person voice, e.g. "we", except in introduction and conclusion.

Response 6: we have rephrased the text in order to avoid the use of first person voice

Comment 7: The text in the figures is difficult to read and needs to be in a darker font.

Response 7: In accordance with the reviewer’s suggestion, figures have been revised using the bold font for the title, increasing the font size and using the black font for all the characters

Reviewer 2 Report

Comments and Suggestions for Authors

This study provides curious insights into the potential effects of cereal biofortification and dietary shifts on iron intake in response to climate-induced nutrient degradation.

The study focuses primarily on iron intake from wheat and rice, leaving out other potential dietary strategies and food sources that could contribute significantly to iron intake. A broader dietary intervention strategy, incorporating additional food sources high in bioavailable iron (such as legumes, leafy greens, or animal products), might offer more effective results, particularly given that bioavailability of plant-based iron is lower than that of heme iron from animal sources.

While the study acknowledges the role of diet composition in iron bioavailability, it lacks a detailed analysis of factors affecting absorption. For example, whole grains contain phytates, which can inhibit iron absorption, potentially counteracting the benefits of increased whole wheat consumption. Similarly, pairing iron sources with foods rich in vitamin C, which enhances absorption, could be an effective dietary approach worth considering in the scenarios.

The scenarios assume that shifts to whole wheat and brown rice consumption are feasible at a population level. However, such shifts may face cultural, economic, and availability barriers. A more in-depth exploration of the feasibility and acceptability of these dietary changes among different demographic groups would strengthen the conclusions.

The manuscript relies heavily on biofortification, yet it does not explore other iron fortification techniques, such as the addition of iron to processed foods or dietary supplements. These methods might offer a more immediate and reliable means of addressing iron deficiency than agricultural biofortification, which is a longer-term solution with variable success depending on regional agricultural conditions.

The analysis would benefit from a sensitivity analysis to test the robustness of the scenarios against various assumptions, such as changes in dietary preferences or economic shifts that might affect food consumption patterns. Climate change may also impact the availability and cost of food, affecting access to iron-rich foods.

The analysis notes that children and women of childbearing age are at higher risk for iron deficiency but does not go into sufficient depth on tailored interventions for these groups. The inclusion of targeted interventions or supplementary measures for these high-risk groups would enhance the relevance and application of the findings.

Author Response

Comment 1: This study provides curious insights into the potential effects of cereal biofortification and dietary shifts on iron intake in response to climate-induced nutrient degradation.

The study focuses primarily on iron intake from wheat and rice, leaving out other potential dietary strategies and food sources that could contribute significantly to iron intake. A broader dietary intervention strategy, incorporating additional food sources high in bioavailable iron (such as legumes, leafy greens, or animal products), might offer more effective results, particularly given that bioavailability of plant-based iron is lower than that of heme iron from animal sources.

Response 1: we thank the reviewer for this comment. We do agree that the inclusion of additional food sources could have been of interest. However, as mentioned in the title and in the text, our intention was to perform a case study with cereals which represent staple foods in the Italian diet and the which iron content could be largely affected by climate change. We have further implemented the discussion adding the point arisen by the reviewer as a potential limitation of the study

Comment 2: While the study acknowledges the role of diet composition in iron bioavailability, it lacks a detailed analysis of factors affecting absorption. For example, whole grains contain phytates, which can inhibit iron absorption, potentially counteracting the benefits of increased whole wheat consumption. Similarly, pairing iron sources with foods rich in vitamin C, which enhances absorption, could be an effective dietary approach worth considering in the scenarios.

Response 2: We do agree with the reviewer’s comment. However, again, we would like to emphasize that we use the biofortification with iron as a case study well conscious that the effective iron intake and iron status might be affected by many factors. Based on this suggestion, we have implemented the introduction and the discussion sections highlighting this point.

Comment 3: The scenarios assume that shifts to whole wheat and brown rice consumption are feasible at a population level. However, such shifts may face cultural, economic, and availability barriers. A more in-depth exploration of the feasibility and acceptability of these dietary changes among different demographic groups would strengthen the conclusions.

Response 3: We thank the reviewer for suggesting this point. Feasibility and acceptability certainly represent a crucial point, and we have commented on this point in the conclusion section.

Comment 4: The manuscript relies heavily on biofortification, yet it does not explore other iron fortification techniques, such as the addition of iron to processed foods or dietary supplements. These methods might offer a more immediate and reliable means of addressing iron deficiency than agricultural biofortification, which is a longer-term solution with variable success depending on regional agricultural conditions.

Response 4: we fully agree that biofortification is only one of the feasible techniques to increase iron in foods. However, as demonstrated in literature, biofortification has been shown to be a cost-effective process since it is a one-time investment to develop bio-fortified crop and recurrent cost are low, differently from fortification techniques (see https://doi.org/10.1016/B978-0-12-818444-8.00010-9). To better clarify this point, we have implemented the Introduction

Comment 5: The analysis would benefit from a sensitivity analysis to test the robustness of the scenarios against various assumptions, such as changes in dietary preferences or economic shifts that might affect food consumption patterns. Climate change may also impact the availability and cost of food, affecting access to iron-rich foods.

Response 5: We thank the reviewer for suggesting this point. The sensitivity of the results presented to the change in this data is straightforward as it is a simple projection: (for example if wheat consumption falls by 10% (on average) we lose 10% of all the projections made). We have added a sentence regarding the sensitivity analysis in the discussion.

Comment 6: The analysis notes that children and women of childbearing age are at higher risk for iron deficiency but does not go into sufficient depth on tailored interventions for these groups. The inclusion of targeted interventions or supplementary measures for these high-risk groups would enhance the relevance and application of the findings.

Response 6: We thank the reviewer for the suggestion. Other strategies of intervention for children and women in fertile age have been added to the discussion.

Reviewer 3 Report

Comments and Suggestions for Authors

After careful consideration, I fell that the manuscript entitled “Simulation of Daily Iron Intake by Actual Diet considering Future Trends in Wheat and Rice Biofortification, Environmental and Dietary Factors: An Italian Case Study” presents some problems. In general, the methods and results presented are not properly presented. Some results in the text do not match those presented in the tables and the poor quality of the pictures do not allow them to be evaluated. In addition, the study presents only a descriptive result. In addition, statistical tests should be carried out to confirm the significance of the observed results.

Some of these problems are listed below.
1. line 176. Did this work simulate 6 or 4 scenarios? Correct this sentence.

2. Section 2.1: The authors say that this study “…cover all four main geographical areas and all classes of age” (line 430). It is therefore important to provide a description of the composition of the sample in these respects. What is the percentage of the sample in each of the 4 regions analyzed? Is this sample representative of the population in these geographical areas?

3. Line 214 and the whole text of the manuscript: replace "/die" by "/day".

4. Statistical Analysis section. The study presents only a descriptive result. Statistical tests should be carried out to confirm the significance of the observed results (see comments below).

5. Line 266: Some results presented in the text do not match those in the tables. For example, it is not clear from which part of table 1 the statement that the median dietary iron intake ranged from 5.8 to 12.8 mg/day.

6. Line 266 the whole text of the manuscript. The statements made do not make sense. What it means is that the median is between P5 and P95. In fact, P5-P95 is not a confidence interval for the median. Similar cases occur in the other parts of the manuscript and should be corrected.

7. Table 1: Why was median (and P5-P95) information not included for “Higher/Lower average iron intake”?

8. Line 284: The authors state that “The mean intake of wheat was highest in the elderly group…”. Is this difference significant? A statistical test should be carried out to verify the significance of these statements (see comment 4).

9. Line 294-297: Rephrase that sentence, as it doesn't seem to make sense. The use of the term “respectively” is confusing. In addition, it is incorrect to say that the median ranged between X and Y (see comment 6).

10. Line 305. The values mentioned do not correspond to those shown in Table 3 (see also comment 6).

11. Line 323: Is this a significant improvement? (see comment 4)

12. Figures 1-4. The figures present poor quality, prejudicing their evaluation. They should be replaced with higher quality ones. Also, it is not clear whether they represent the mean or the median. In addition, figures could accompany the error bars with SD or percentile intervals.

13. Table A1. It is not clear whether the values represent the mean or the median. Use “.” to separate decimals.

Author Response

After careful consideration, I fell that the manuscript entitled “Simulation of Daily Iron Intake by Actual Diet considering Future Trends in Wheat and Rice Biofortification, Environmental and Dietary Factors: An Italian Case Study” presents some problems. In general, the methods and results presented are not properly presented. Some results in the text do not match those presented in the tables and the poor quality of the pictures do not allow them to be evaluated. In addition, the study presents only a descriptive result. In addition, statistical tests should be carried out to confirm the significance of the observed results.
Some of these problems are listed below.

Comment 1: 1. line 176. Did this work simulate 6 or 4 scenarios? Correct this sentence.
Response 1: We are sorry for the typo that has been modified. Iron intake and adequacy estimation trends were categorized in four scenarios compared to a baseline, for a total of 5 scenarios.

Comment 2: Section 2.1: The authors say that this study “…cover all four main geographical areas and all classes of age” (line 430). It is therefore important to provide a description of the composition of the sample in these respects. What is the percentage of the sample in each of the 4 regions analyzed? Is this sample representative of the population in these geographical areas?
Response 2: We thank the reviewer for the comment. The IV SCAI 2017-2020 data were collected in over 800 children aged between 3 months and 9 years, and approximately 1,200 adolescents, adults, and older adults aged 10-74 years, stratified also by sex and geographical areas (Northwest, Northeast, Central, and Southern Italy, including islands). We have added these details in section 2.1

Comment 3: Line 214 and the whole text of the manuscript: replace "/die" by "/day".
Response 3: Done as suggested

Comment 4: Statistical Analysis section. The study presents only a descriptive result. Statistical tests should be carried out to confirm the significance of the observed results (see comments below).
Response 4: We thank the reviewer for the comment. The Kruskal Wallis test was used to compare mean intake values of wheat and rice variables. We added these details in the statistical analyses section and in the results.

Comment 5: Line 266: Some results presented in the text do not match those in the tables. For example, it is not clear from which part of table 1 the statement that the median dietary iron intake ranged from 5.8 to 12.8 mg/day.
Response 5: Table 1 reports data related to iron intake in males, females and related age groups, and in all group, expressed both as mean and median. For instance, the sentence “Median dietary iron intake ranged from 5.8 to 12.8” means that among males the lowest median was 5.8 mg/day (in the 1-2 years group) to 12.8 (in the 65-75 y group”. We have revised section 3.1 to improve clarity

Comment 6: Line 266 the whole text of the manuscript. The statements made do not make sense. What it means is that the median is between P5 and P95. In fact, P5-P95 is not a confidence interval for the median. Similar cases occur in the other parts of the manuscript and should be corrected.
Response 6: We thank the reviewer for noting that this section was not clear. As mentioned above, 5.8 and 12.8 mg/day do not represent P5 and P95 but the lowest and the highest median found in the different age groups. (1-2 y and 65-74 y age groups, respectively).

Comment 7: Table 1: Why was median (and P5-P95) information not included for “Higher/Lower average iron intake”?
Response 7: Thank you for your comment. The median with the percentiles is included now in Table 1.

Comment 8: Line 284: The authors state that “The mean intake of wheat was highest in the elderly group…”. Is this difference significant? A statistical test should be carried out to verify the significance of these statements (see comment 4).
Response 8: We thank the reviewer for the comment.  We added statistical details in the results section.

Comment 9: Line 294-297: Rephrase that sentence, as it doesn't seem to make sense. The use of the term “respectively” is confusing. In addition, it is incorrect to say that the median ranged between X and Y (see comment 6).
Response 9: Here we referred to the median in the different target group. In this case “The median iron intake from wheat and rice consumption ranged from 0.4 to 1.8 mg/day” means that the lowest median in males was observed in the <1 y age group (i.e. 0.4) and the highest in the 10-18 and 65-75 age groups (1.8).

Comment 10: Line 305. The values mentioned do not correspond to those shown in Table 3 (see also comment 6).
Response 10: 199.6 8106.4) corresponds to the mean cereal intake in the whole group, as reported in the third column of Table 3. We have revised the sentence to increase the ease of reading.

Comment 11. Line 323: Is this a significant improvement? (see comment 4)
Response 11: We thank the reviewer for the comment. We added Table 4 with statistical details in the results section on iron improvement.

Comment 12: Figures 1-4. The figures present poor quality, prejudicing their evaluation. They should be replaced with higher quality ones. Also, it is not clear whether they represent the mean or the median. In addition, figures could accompany the error bars with SD or percentile intervals.
Response 12: Quality of figures have been improved.

Comment 13. Table A1. It is not clear whether the values represent the mean or the median. Use “.” to separate decimals.

Response 13: Done as suggested, thank you.

Reviewer 4 Report

Comments and Suggestions for Authors

The article entitled “Simulation of daily iron intake according to the real diet considering future trends in wheat and rice biofortification, environmental and dietary factors: an Italian case study” has an essentially innovative character worthy of being taken into account, in addition to having a very interesting content. Its location in Italy is the first step to be able to disseminate the study, through this publication, to other countries, both European and in other contexts where these crops are a priority. Biofortification, as an idea to improve crops and increase their nutritional value, is a fact of vital importance for the development of some areas of humanity.

The authors present a broad and well-founded introduction that ends by setting the objective: to quantify in the current Italian diet and in different age groups the contribution of the cereal food group (wheat and rice) to the daily iron intake and to compare the total intake with the recommended level.

To achieve the objective, the method simulates 4 scenarios of iron content trends in wheat and rice over the next 20 years, considering biofortification, the environment, and mental and dietary factors. The questionnaire used, as well as the accompanying graphs and appendix, are well-prepared and help to understand the study.

The work is accompanied by 76 references appropriate to the study, most of them very recent.

The discussion is well-founded, based on the authors cited in the introduction, which makes the results more credible, and the conclusions respond to the objective, also raising future perspectives.

In general, it is a good article. I only find that in line 176 they talk about 6 scenarios, but in the methodology they only raise 4. It must be a mistake, please modify it.

Author Response

Comment 1: The discussion is well-founded, based on the authors cited in the introduction, which makes the results more credible, and the conclusions respond to the objective, also raising future perspectives.

Response 1: We thank the reviewer for the positive comments about our work.

Comment 2: In general, it is a good article. I only find that in line 176 they talk about 6 scenarios, but in the methodology, they only raise 4. It must be a mistake, please modify it.

Response 2: The typo has been modified, thank you for noting it. Iron intake and adequacy estimation trends were categorized in four scenarios compared to a baseline, for a total of 5 scenarios.

Round 2

Reviewer 3 Report

Comments and Suggestions for Authors

I believe that the authors have made the corrections and/or provided the justifications satisfactorily. However, I still feel that the quality of the figures is not suitable for publication.